# OmniGen-AR: AutoRegressive Any-to-Image Generation

**Junke Wang**[1,2]**, Xun Wang**[3]**, Qiushan Guo**[3]**, Peize Sun**[4]**,**
**Weilin Huang**[3]**, Zuxuan Wu**[1,2†]**, Yu-Gang Jiang**[1,2]
[1]Institute of Trustworthy Embodied AI, Fudan University
[2]Shanghai Collaborative Innovation Center of Intelligent Visual Computing
[3]Bytedance Seed, [4]The University of Hong Kong

## Abstract

Autoregressive (AR) models have demonstrated strong potential in visual generation, offering superior performance with simple architectures and optimization objectives. However, existing methods are typically limited to single-modality conditions, *e.g.*, text, restricting their applicability in real-world scenarios that demand image synthesis from diverse controls. In this work, we present OmniGen-AR, a unified autoregressive framework for Any-to-Image generation. By discretizing various visual conditions through a shared visual tokenizer and text prompts with a text tokenizer, OmniGen-AR supports a broad spectrum of conditional inputs within a single model, including text (text-to-image generation), spatial signals (segmentation-to-image and depth-to-image), and visual context (image editing, frame prediction, and text-to-video generation). To mitigate the risk of information leakage from condition tokens to content tokens, we introduce Disentangled Causal Attention (DCA), which separates the full-sequence causal mask into condition causal attention and content causal attention. It serves as a training-time regularizer without affecting the standard next-token prediction during inference. With this design, OmniGen-AR achieves new state-of-the-art or at least competitive results across a range of benchmark, *e.g.*, 0.63 on GenEval and 80.02 on VBench, demonstrating its effectiveness in flexible and high-fidelity visual generation.

## 1 Introduction

In recent years, deep generative models [32, 21, 100, 61, 52] have experienced rapid development and revolutionized the way we create visual contents. Among them, autoregressive models (AR) [15, 100, 66, 82] have demonstrated the capability for high-quality image synthesis through sequential token prediction. The superior performance, flexbility, and compatibility with multimodal inputs, position them as competitive alternatives to diffusion models [25, 64, 13, 51, 70].

Despite their potential, existing autoregressive (AR) visual generation methods primarily focus on single-modality conditioning, such as category labels [15, 18, 101, 81, 79] or text prompts [60, 100, 66, 82]. While these models achieve strong performance within the respective domains, they fall short of the versatility required in real-world applications, where visual generation often respond to a diverse set of conditional inputs [106, 107, 89, 94], such as semantic masks, reference images, or history frames. In other words, building a unified AR generative model that accommodates various inputs remains under-explored.

To fill this gap, this work presents OmniGen-AR, an autoregressive framework for **Any-to-Image generation**. In addition to text, OmniGen-AR also supports a wide range of visual conditions

---

†: corresponding author.

39th Conference on Neural Information Processing Systems (NeurIPS 2025).

Table 1: A system-level comparison between OmniGen-AR and other methods. Compared to OmniGen [94], OmniGen-AR additionally supports video generation.

| Method | Type | Condition | | |
| | | Text | Ref | Spatial |
| --- | --- | --- | --- | --- |
| GLIGEN [40] | Diff | ✓ | ✗ | ✓ |
| ControlNet [106] | Diff | ✓ | ✗ | ✓ |
| Uni-ControlNet [107] | Diff | ✓ | ✗ | ✓ |
| OmniGen [94] | Diff | ✓ | ✓ | ✓ |
| LLamaGen [66] | AR | ✓ | ✗ | ✗ |
| SimpleAR [82] | AR | ✓ | ✗ | ✗ |
| ControlAR [41] | AR | ✓ | ✗ | ✓ |
| Ours | AR | ✓ | ✓ | ✓ |

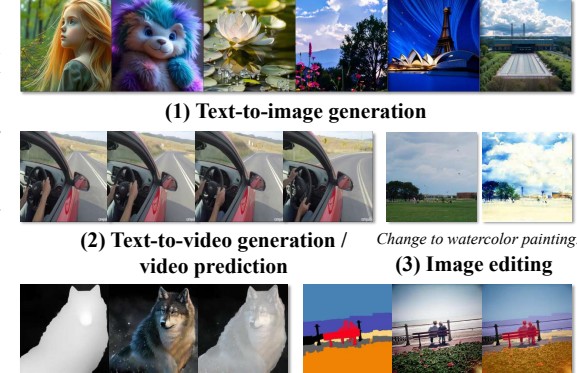

(1) Text-to-image generation

(2) Text-to-video generation / video prediction

*Change to watercolor painting.*
(3) Image editing

(4) Depth-to-image generation

(5) Seg-to-image generation

Figure 1: OmniGen-AR could handle 5 types of generation tasks within a single model.

including segmentation masks, depth maps, and reference images, by discretizing them with a shared visual tokenizer. An overview of our method is provided in Table 1 and Figure 1. While this approach allows our model to preserve the simplicity of autoregressive modeling, the serial nature of prediction introduces a potential risk of information leakage from condition tokens to content tokens. This becomes particularly problematic in tasks like image editing and frame prediction, where much of the output remains unchanged relative to the input. Consequently, the model may converge to suboptimal solutions that exploit shortcut patterns between conditioning and prediction signals, instead of producing meaningful and instruction-following results.

To address this, we introduce Disentangled Causal Attention (DCA), which separates the causal attention over the entire sequence into *condition causal attention* and *content causal attention*. This disentanglement prevents the information flow from content tokens to condition tokens while still allowing the latter to retain awareness of their relative positions. During training, we randomly replace the vanilla causal attention with DCA, which regularizes the model by discouraging over-reliance on conditional context and promoting more instruction-compliant predictions. The inference process of our model still follows the standard next-token prediction.

We validate the effectiveness of OmniGen-AR on six representative visual generation tasks, including text-to-image generation, text-to-video generation, frame prediction, image editing, depth-conditioned image generation, and segmentation-conditioned image generation. The results demonstrate that it achieves new state-of-the-art or at least competitive results on the prevalent benchmarks, *e.g.*, 0.63 on GenEval [20] and 80.02 on VBench [29]. OmniGen-AR not only maintains the inherent flexibility and scalability of autoregressive models but also enables the seamless integration of various control signals, providing a unified and effective solution for universal visual generation.

## 2 Related Work

### 2.1 Autoregressive Visual Generation

Autoregressive (AR) models have become a popular paradigm in generative modeling for both language [56, 57, 72, 3] and vision [100, 66, 82], owing to their strong capability in modeling complex distributions. Early efforts in AR-based visual generation model images as sequences of pixels [74, 10], which achieves satisfactory results but suffer from inefficiency and limited scalability. Subsequent approaches such as VQ-VAE [75] introduce discrete visual tokenization to autoregressive visual generation, enabling the use of transformer-based language models for image synthesis. These token-based methods significantly improve the generation quality and training stability, attracting a series of work that leverage learned codebooks for autoregressive image generation [60, 15, 100, 66].

Recent works have explored autoregressive generation with continuous representations [39, 108], scale-wise autoregressive modeling [69, 23], and reinforcement-learning for improved generation quality [82]. Despite these advances, existing AR models mainly focus on single-modality conditions

(*e.g.*, text or class labels), restricting their applicability in real-world scenarios requiring multi-modal controls. The most relevant literature is EditAR [49], which also employs autoregressive transformers and support multiple conditional image generation tasks. Ours work differ from them in two aspects: 1) EditAR is specifically designed for image editing and low-level control tasks (*e.g.*, depth-to-image, edge-to-image, segmentation-to-image), while our model is a unified Any-to-Image framework that handles a broader range of input modalities. 2) EditAR aims to improve text-image alignment by introducing distillation loss, while OmniGen-AR hopes to prevent information leakage through DCA, a novel training-time attention mechanism that disentangles condition and content attention paths.

## 2.2 Diffusion Models for Any-to-Image Generation

Recent advances in diffusion models [52, 14] have significantly improved the quality and controllability of image synthesis from diverse conditioning signals [106, 40, 85, 104]. ControlNet [106] firstly introduces a framework that injects spatial control (*e.g.*, edge maps, segmentation masks) into a pretrained diffusion model without compromising generation quality. It demonstrates strong performance in aligning generated content with spatial priors but still requires separate adapters for each conditional modality. To address this, Uni-ControlNet [107] proposes a unified architecture that supports multiple spatial controls within a single framework by learning a modality-agnostic representation space. It improves the generality and flexibility across tasks, but still requires the separate training procedures for different types of condition. More recently, OmniGen [94] pushes toward general-purpose image generation by unifying diverse tasks, *e.g.*, text-to-image synthesis, image editing, and subject-driven generation, within a diffusion framework that eliminates external modules. Inspired by this, we explore the autoregressive framework for any-to-image generation. We adopt autoregressive modeling as it provides a more natural fit for handling sequential inputs and enabling interleaving generation.

## 2.3 Unified Models for Multimodal Understanding and Generation

The belief in scaling data and model size [24, 31, 102] has driven the community towards building unified and even general multimodal models [2, 68, 1]. CLIP [55] first demonstrates that large-scale contrastive pretraining on image-text pairs could yield powerful vision-language representations. Subsequent works [38, 80, 88, 12, 4] extend this paradigm to support a broader range of vision-language tasks such as captioning and VQA, across both image and video domains.

LLaVA [44, 43] opens up another chapter, *i.e.*, visual instruction tuning, by aligning pretrained vision encoders [57, 103] with large language models [72] to enabling open-ended multimodal understanding. Recently, Chameleon [67] steps beyond the scope of understanding tasks to unified multimodal understanding and generation, seamlessly integrating both modalities in a token-based framework. Following work improve the design of unified multimodal language models through better multimodal fusion [111, 95, 71] and visual encoding [92, 93]. These advancements showcase the growing potential of general multimodal artificial intelligence, pushing the boundaries of both understanding and generation tasks across multiple domains.

# 3 Method

Our goal is to unify conditional image generation (*i.e.*, text, spatial, image) within a single autoregressive framework. To this end, we propose OmniGen-AR, which consists of a text and visual tokenizer to discrete various inputs. With this, we model the dependency between multimodal tokens using an autoregressive transformer. The architecture of OmniGen-AR is illustrated in Figure 2.

## 3.1 Visual and Textual Tokenization

To enable the unified processing and generation of diverse modalities, the first question is how to represent them in a compatible format. Unlike previous work [106, 41] that rely on separate encoders to encode visual condition $V \in \mathbb{R}^{H \times W \times 3}$ (segmentation masks, depth maps, image to be edited) and the image to be generated $X \in \mathbb{R}^{H \times W \times 3}$, we adopt the same visual tokenizer [16] to convert them into discrete visual tokens: $v \in \mathbb{R}^{N_1}$ and $x \in \mathbb{R}^{N_2}$, where $N1 = N2$ denote the sequence length. Here we omit the loop-up and flatten operations for simplicity. While for the textual inputs, we tokenize them with a language model tokenizer [98] to obtain the text tokens $t \in \mathbb{R}^M$.

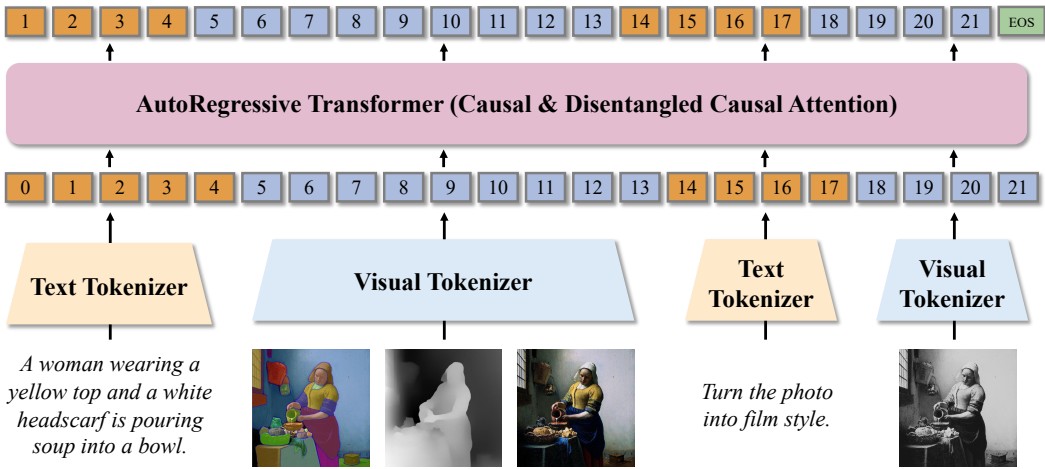

Figure 2: OmniGen-AR consists of a text tokenizer, a visual tokenizer, an autoregressive transformer.

## 3.2 Autoregressive Transformer for Multimodal Generation

We adopt a decoder-only transformer model for multimodal generation, which consists of stacked attention blocks [76]:

$$\text{Attention}(q, k, v) = \text{softmax}\left(\frac{qk^{\text{T}}}{\sqrt{d_k}} + m\right) v, \tag{1}$$

where $q$, $k$, $v \in \mathbb{R}^{N \times d_k}$ represent the query, key, and value embeddings, and $m \in \mathbb{R}^{N \times N}$ is the attention mask. In modern language models [57, 72] and AR-based visual generation models [100, 66, 82], $m$ is usually implemented as a lower triangular matrix to mask out the future positions :

$$m_{i,j} = \begin{cases} 0, & \text{if } j \leq i \\ -\infty, & \text{otherwise} \end{cases} \tag{2}$$

This mechanism ensures that each token attends only to itself and preceding tokens in the sequence, preserving the left-to-right generation order. However, in the context of conditional image generation, the plain causal attention can lead to unintended information leakage: given access to previous condition tokens, the model may learn to exploit trivial correlations between them and the content tokens to be predicted, instead of generating meaningful tokens that follow instructions faithfully [17, 96, 37, 84].

To alleviate this problem, we modify the attention mask to prevent information flow from condition tokens to content tokens, while still preserving autoregressive modeling within each. Taking the image editing task as an example, we concatenate text, condition, and content tokens along the sequence dimension as the token sequence: $[t, v, x] \in \mathbb{R}^L$, where $L = M + N_1 + N_2$, and design the attention mask $m \in \mathbb{R}^{(M+N_1+N_2) \times (M+N_1+N_2)}$ in the following manner:

$$m_{i,j} = \begin{cases} 0, & \text{if } j \leq i \text{ and } (i,j) \in A \cup B \cup C \\ -\infty, & \text{if } i \in C, j \in B \\ -\infty, & \text{otherwise} \end{cases} \tag{3}$$
$$\textbf{where } A = [0, M), \quad B = [M, M+N_1), \quad C = [M+N_1, M+N_1+N_2)$$

Such a masking scheme permits content tokens to attend to preceding text tokens while blocking access to other condition tokens, thereby reducing the risk of shortcut learning. During training, we randomly apply DCA in place of vanilla causal attention as a regularization. Please see Figure 3 for a better illustration. Notably, the proposed DCA differs from classifier-free guidance [26] in its treatment of condition tokens. First of all, content tokens remain aware of positional information, as the condition tokens are not dropped entirely. In addition, DCA is applied only during training and has no impact on the inference process.

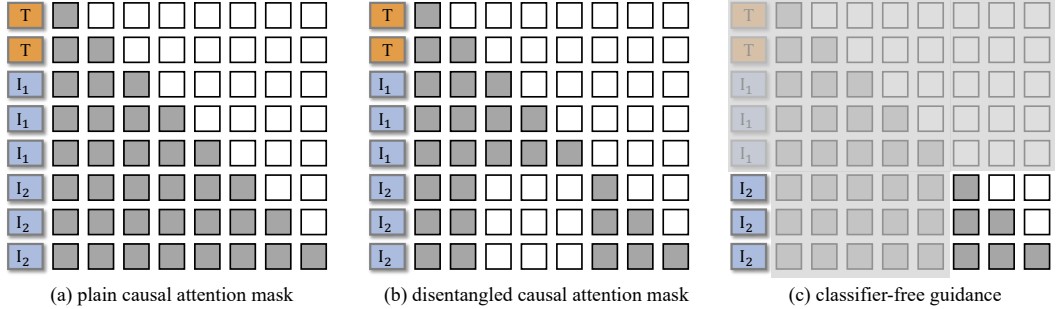

|  (a) plain causal attention mask | (b) disentangled causal attention mask | (c) classifier-free guidance |

Figure 3: Comparison between plain causal attention, the proposed disentangled causal attention, and classfier-free guidance. T, $I_1$, $I_2$ represent text, spatial (image) condition and content tokens.

## 3.3 Training and Inference

For different generation tasks, we construct the token sequence $z$ by interleaving text, condition, and content tokens according to task-specific formats. Specifically, spatial- and image-conditioned tasks follow the manner: $z = [t, v, x]$, while text-conditioned tasks use $z = [t, x]$. OmniGen-AR is trained to autoregressively predict the next token over these sequences using language modeling loss [57, 100].

During inference, the tokens are sequentially sampled based on the learned conditional probability: $\hat{z}_i = \arg\max p_\theta(z_i|z_{<i})$. After that, we feed them to the decoder of visual tokenizer to generate images. Classfier-Free Guidance (CFG) [26] is adopted to improve the generation quality, following previous work [66, 82].

## 4 Experiments

### 4.1 Experimental Setup

**Training data.** The training of OmniGen-AR includes three stages: 1) single image stage (SI), where we pretrain our model on large-scale image datasets, involving CC3M [62], CC12M [7], OpenImages [36], SAM1B [33], and Megalith-huggingface [46]. We also incorporate video datasets, *i.e.*, a 9M subset of Panda70M [11] and HD-VILA-100M [97], and randomly sample 1 frame for each video. 2) image-video joint stage (IV), where we maintain the datasets used in the first stage but sampled 9 frames from the videos. 3) multi-task stage (MT), where we train our model on a wide-range of high-quality datasets, including text-to-image datasets (JourneyDB [65], Synthetic-dataset-1M [53], and 10M internal data), image editing datasets (MagicBrush [105], Instruct-Pix2Pix [5], SEED-Edit [19]), depth-to-image datasets (MultiGen-Depth [54]), segmentation-to-image datasets (MultiGen-ADE20k [54] and MultiGen-COCOStuff [54]), and text-to-video datasets (OpenSora-pexels-45k [28], OpenVid-1M [50], and 0.5M high-quality internal data). We recaption all the images and videos using Qwen2-VL [83].

**Implementation details.** We adopt Qwen2.5 [98] as the text tokenier and transformer model. While for visual tokenizer, we use an image-video joint tokenizer, *i.e.*, Cosmos-DV8$\times$16$\times$16 [16], which allows us to tokenize different controls, images, and videos with the same codebook. During the SI and IV stages, we train our model on 512 resolution, and the learning rate is set to 1e-4. While for the MT stage, we increase the resolution to 1024 and decrease the learning rate to 2e-5. We train our model on 64 A100 GPUs, the global batch size is 256 for all stages, no warm up or learning rate decay are used. AdamW [45] is employed for optimization. During the IV and MT stages, we replace the standard causal attention mask with a disentangled causal attention mask with a probability of 10%, and similarly drop the text conditions for classifier-free guidance with the same probability. We set CFG scale to 6.0 during inference.

### 4.2 Comparison with State-of-the-arts

**Text-to-image generation.** In Table 2, we compare OmniGen-AR with existing image generation models on GenEval [20], a challenging and popular text-to-image (T2I) benchmark. The results

Table 2: Text-to-image generation on GenEval, Results markdd with † result are using prompt rewriting.

| Method | Par. | Two. | Pos. | Color. | Overall |
|---|---|---|---|---|---|
| SDv1.5 [61] | 0.9B | 0.38 | 0.04 | 0.06 | 0.43 |
| PixArt-alpha [9] | 0.6B | 0.50 | 0.08 | 0.07 | 0.48 |
| SDv2.1 [61] | 0.9B | 0.51 | 0.07 | 0.17 | 0.50 |
| LlamaGen [66] | 0.8B | 0.34 | 0.07 | 0.04 | 0.32 |
| SimAR-SFT [82] | 0.5B | **0.75** | 0.20 | 0.24 | 0.53 |
| Ours | 0.5B | 0.74 | **0.20** | **0.29** | **0.55** |
| LDM [61] | 1.4B | 0.29 | 0.02 | 0.05 | 0.37 |
| DALL-E 2 [59] | 6.5B | 0.66 | 0.10 | 0.19 | 0.52 |
| Show-o [95] | 1.3B | 0.80 | 0.31 | 0.50 | 0.68 |
| Infinity [23] | 2B | $0.85^\dagger$ | $\mathbf{0.49}^\dagger$ | $\mathbf{0.57}^\dagger$ | $\mathbf{0.73}^\dagger$ |
| Janus [92] | 1.5B | 0.68 | 0.46 | 0.42 | 0.61 |
| Emu3 [86] | 8.5B | $0.81^\dagger$ | $\mathbf{0.49}^\dagger$ | $0.45^\dagger$ | $0.66^\dagger$ |
| SimAR-SFT [82] | 1.5B | 0.87 | 0.27 | 0.33 | 0.61 |
| Ours | 1.5B | **0.94** | 0.30 | 0.40 | 0.63 |

Table 3: Video generation on VBench.

| Method | Par. | Qua. | Sem. | Total |
|---|---|---|---|---|
| CogVideo [27] | 9B | 72.06 | 46.83 | 67.01 |
| LaVie [87] | 3B | 78.78 | 70.31 | 77.08 |
| OpSoraP V1.3 [42] | 2.7B | 80.14 | 65.62 | 77.23 |
| CogVideoX [99] | 5B | 83.05 | 77.33 | 81.91 |
| Hunyuan [35] | 13B | 85.09 | 75.82 | 83.24 |
| Mira [30] | 1.1B | 78.78 | 44.21 | 71.87 |
| TF-T2V [8] | 1.8B | 80.05 | 56.69 | 75.38 |
| OpSora V1.2 [109] | 1.1B | 81.35 | 73.39 | 79.76 |
| AniDiff V2 [22] | 0.9B | 82.90 | 69.75 | 80.27 |
| VidCrafter-2.0 [8] | 1.4B | 82.20 | 73.42 | 80.44 |
| CogVideoX [99] | 2B | 82.18 | 75.83 | 80.91 |
| Wan2.1 [78] | 1.3B | **85.23** | 75.65 | **83.31** |
| Ours | 0.5B | 76.60 | 67.20 | 74.72 |
| Ours | 1.5B | 81.51 | **78.08** | 80.02 |

Table 4: Frame prediction on Kinetics-600 (left, $^*$ denotes zero-shot evaluation), image editing on Emu-Edit test set (middle), and spatial-conditioned generation (right). CT: CLIP text similarity between edited image and edited prompt, CI: CLIP image similarity between edited image and condition image.

| Method | FVD (↓) |
|---|---|
| LVT [58] | 225 |
| ViTrans [91] | 170 |
| CogVideo [27] | 109 |
| ViVQVAE [77] | 64 |
| OmniTok [81] | **33** |
| VideoPoet-8B$^*$ [34] | 687 |
| Ours-1.5B$^*$ | 429 |

| Method | CT | CI |
|---|---|---|
| I-Pix2Pix [5] | 0.22 | 0.83 |
| MagBrush [105] | 0.22 | 0.84 |
| PnP [73] | 0.09 | 0.52 |
| Null-Text [47] | 0.24 | 0.76 |
| Emu-Edit [63] | 0.23 | **0.86** |
| OmniGen [94] | 0.23 | 0.83 |
| Ours-1.5B | **0.23** | 0.84 |

| Method | Mask mIoU (↑) | Depth RMSE (↓) |
|---|---|---|
| Uni-ControlNet [107] | 19.39 | 40.65 |
| GLIGEN [40] | 23.78 | 38.83 |
| EditAR [49] | 22.62 | 34.93 |
| ControlNet [106] | 32.55 | 35.90 |
| ControlAR [41] | 39.95 | **29.01** |
| OmniGen [94] | **40.06** | 31.71 |
| Ours-1.5B | 35.28 | 37.42 |

show that OmniGen-AR significantly outperforms all other models with fewer than 1B parameters, including both diffusion models (*e.g.*, SDv2.1 [61]) and autoregressive models (*e.g.*, LLamaGen [66]). When scaled to 1.5B size, the overall performance of OmniGen-AR is improved from 0.57 to 0.63, highlighting its promising scalability when more training computes are available.

**Text-to-video generation.** We also evaluate OmniGen-AR on VBench [29] for text-to-video generation, and the comparison with existing video generation models is shown in Table 3. With only 0.5B parameters, our model achieves 74.72 total score on VBench, surpassing the previous SOTA AR-based models, *i.e.*, CogVideo [27], by 11% while using much fewer parameters (0.5B *v.s.* 9B). Similar to what we have seen on T2I, the results of T2V could be signigicantly improved to 80.02 using 1.5B parameters, even beating diffusion models like OpenSora V1.2 [109]. It is also worth noting that it is the first time that a vanilla autoregressive model using discrete tokens could achieve 80+ score on VBench.

**Frame prediction and image editing.** To evaluate the image generation capability given visual context (image condition), we choose two types of tasks: frame prediction on Kinetics-600 [6] and image editing on Emu-Edit test set [63]. The results in Table 4 show that OmniGen-AR achieves much lower Fréchet Video Distance (FVD) than VideoPoet [34] for frame prediction. While for the more challenging image editing task, OmniGen-AR also achieves competitive results, *i.e.*, 0.23 CLIP text similarity [55].

**Segmentation and depth-to-image generation.** We follow previous work [41, 94] to report the segmentation-to-image and depth-to-image generation performance on ADE20K [110] and MultiGen-

Depth-Eval [54] in Table 4 (right). Compared to text or image condition, spatial conditions provide more structured and fine-grained instructions, thus posing a greater challenge to the model to geometrically accurate and contextually coherent content. We can see that OmniGen-AR achieves competitive results on both tasks, outperforming diffusion counterparts [48, 40].

## 4.3 Ablation Studies

**Effects of disentangled causal attention.** To better illustrate the potential information leakage in image-conditioned generation tasks, we compute the token match ratio (TMR) of MagicBrush [105], a popular image editing dataset, and visualize it in Figure 4. TMR is defined as the fraction of identical tokens in the same position between condition and content images: $\text{TMR} = \frac{1}{N_1} \sum_i \mathbf{1}\,[v_i = c_i]$, where $v_i$ and $c_i$ denote the $i$-th token from the condition and content images respectively, and $N_1$ is the total number of tokens. The x-axis of Figure 4 denotes binned TMR ranges (*e.g.*, 0.80–0.85), and the y-axis shows the proportion of samples falling into each bin. As can be seen, a significant portion of samples exhibit high TMR values, suggesting substantial overlap between condition and content images, which may imply unintended information leakage.

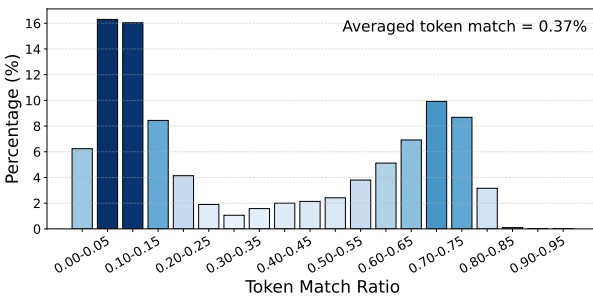

Figure 4: Token similarity on MagicBrush [105].

Table 5: OmniGen-AR w/ and w/o DCA on different generation tasks.

| Method | VBen | Emu-CT | Mask |
|---|---|---|---|
| 0% | 70.33 | 0.15 | 24.76 |
| 5% | 74.55 | 0.17 | 25.78 |
| 10% | 74.72 | 0.20 | 25.33 |
| 20% | 73.28 | 0.21 | 25.16 |
| 30% | 71.69 | 0.19 | 21.49 |

As mentioned in Sec.4.1, we randomly replace standard causal attention with the proposed Disentangled Causal Attention (DCA) during training. We also conduct experiments with varying replacement probabilities using the 0.5B model across different tasks. As shown in Table 5, adopting DCA with a 10% probability improves the CLIP text similarity on Emu from 0.15 to 0.20, indicating the encouragement robust conditioning without significantly limiting the access to informative context. Interestingly, DCA also yields slight gains in segmentation-to-image generation, which we hypothesize results from its ability to reduce over-reliance on exact segmentation inputs and thus improving the robustness of our model. Unless otherwise specified, we adopt a default replacement probability of 10% in all experiments.

Table 6: Joint or separate training.

| Method | GEval | VBen | Emu | Mask |
|---|---|---|---|---|
| Joint | 0.55 | 74.72 | 0.20 | 25.33 |
| T2I | 0.57 | - | - | - |
| T2V | - | 77.18 | - | - |
| Edit | - | - | 0.18 | - |
| Seg | - | - | - | 22.59 |

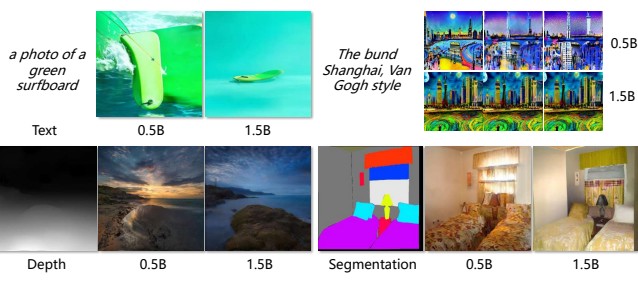

Figure 5: Effects of model scaling.

**Synergy between different tasks.** We study the synergy between different generation tasks by comparing joint training and separate training, both initialized with the 2nd stage checkpoint. As shown in Table 6, joint training leads to degraded performance on text-to-image and text-to-video benchmarks, possibly due to the lower visual quality of editing and spatial-conditioned datasets. In contrast, it improves the results on editing and segmentation-to-image generation tasks, suggesting that strengthening the foundation capability, *i.e.*, text-conditioned generation, can facilitate better generalization to a broader range of downstream tasks.

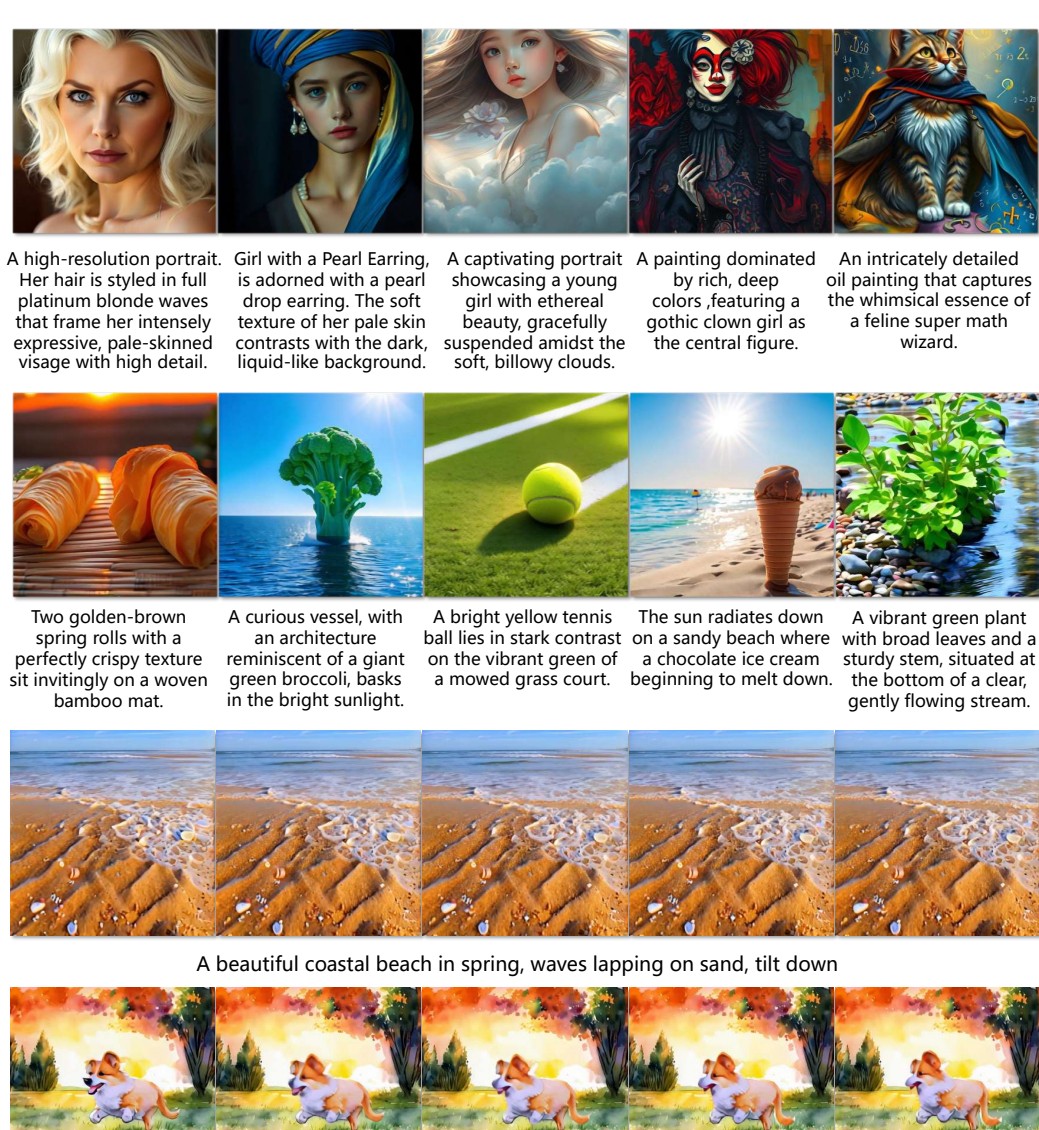

A high-resolution portrait. Her hair is styled in full platinum blonde waves that frame her intensely expressive, pale-skinned visage with high detail.

Girl with a Pearl Earring, is adorned with a pearl drop earring. The soft texture of her pale skin contrasts with the dark, liquid-like background.

A captivating portrait showcasing a young girl with ethereal beauty, gracefully suspended amidst the soft, billowy clouds.

A painting dominated by rich, deep colors ,featuring a gothic clown girl as the central figure.

An intricately detailed oil painting that captures the whimsical essence of a feline super math wizard.

Two golden-brown spring rolls with a perfectly crispy texture sit invitingly on a woven bamboo mat.

A curious vessel, with an architecture reminiscent of a giant green broccoli, basks in the bright sunlight.

A bright yellow tennis ball lies in stark contrast on the vibrant green of a mowed grass court.

The sun radiates down on a sandy beach where a chocolate ice cream beginning to melt down.

A vibrant green plant with broad leaves and a sturdy stem, situated at the bottom of a clear, gently flowing stream.

A beautiful coastal beach in spring, waves lapping on sand, tilt down

A cute happy Corgi playing in park, sunset, watercolor painting

Figure 6: Visualization of text-to-image and text-to-video results generated by OmniGen-AR.

**Model scaling.** In Figure 5, we qualitatively compare the models with 0.5B and 1.5B parameters. It can be seen that scaling the model size could effectively improve the generation results on various tasks, leading to improved instruction-following capability and more aesthetically pleasing images.

## 4.4 Qualitative Results

We display some visualization results in Figure 6 and 7. OmniGen-AR could synthesize high-quality images based on various types of conditions, showcasing both versatility in handling diverse inputs and the ability to maintain semantic coherence with contexts. Several failure cases are also shown in Figure 8, which can be broadly categorized into two types: 1) Instruction-following capability. For instance, in the first row of Figure 8, the instruction is "Remove the bag on the bench next to the person sitting at the bus stop", but the model removes the person instead. This indicates a failure in grounding fine-grained spatial and referential cues from language into visual modifications. 2) Low-quality generations under sparse control signals. Examples in the second row (depth-to-image and segmentation-to-image) show blurry or structurally inconsistent results, which likely stem from noisy supervision and sparse training coverage for these conditions. These failure modes suggest two

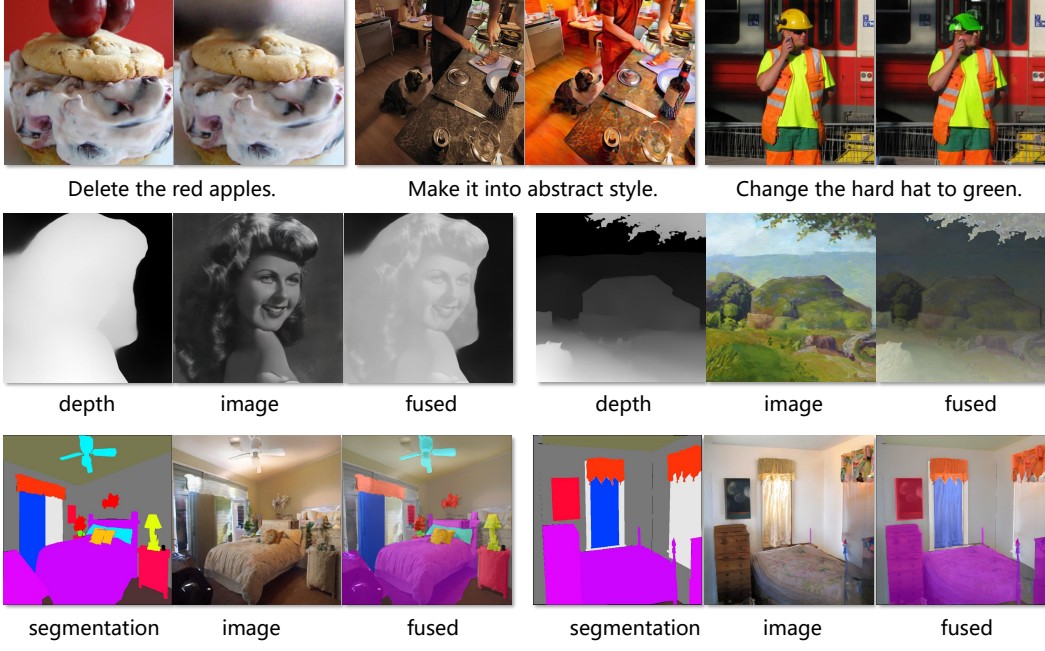

Figure 7: Image editing, depth-to-image, and segmentation-to-image generation results.

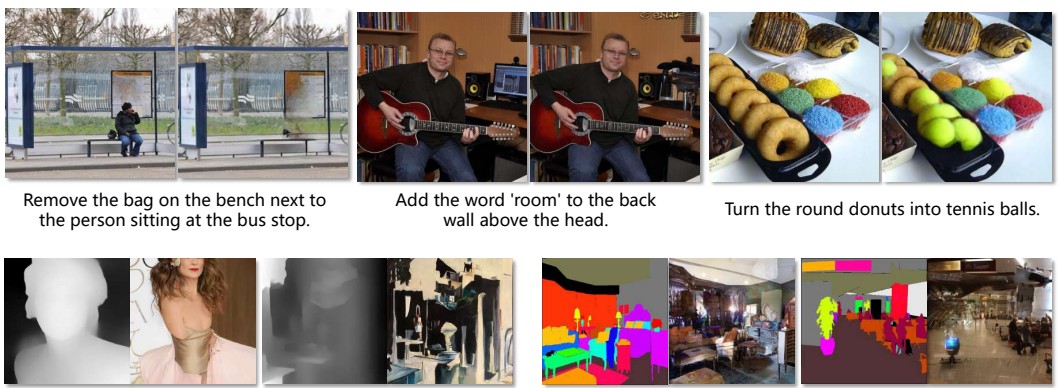

Figure 8: Failure cases generated by OmniGen-AR.

potential directions for future work: 1) Scaling up the model and training data to build a stronger base model with improved generalization and instruction-following ability across diverse visual tasks. 2) Leveraging chain-of-thought (CoT) [90] to improve the reasoning ability on complex prompts.

## 5 Conclusion and Broader Impacts

This paper presented OmniGen-AR, a unified autoregressive framework for any-to-image generation. OmniGen-AR represents a wide spectrum of conditional inputs, *i.e.*, text prompts, spatial controls, and visual contexts, as discrete tokens, and trains a unified autoregressive transformer to model the dependencies between these conditions and the target image tokens. To mitigate the potential information leakage in conditional generation tasks, we proposed Disentangled Causal Attention (DCA), which separates the attention pathways between condition and content tokens to facilitate the learning of instruction-following generation. Comprehensive experiments demonstrate that OmniGen-AR achieves state-of-the-art or at least competitive performance across a wide range of visual generation tasks.

**Acknowledgement** This project was supported by NSFC under Grant No. 62032006 and No. 624B2043.

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
