# OpenReview forum: "OmniGen-AR: AutoRegressive Any-to-Image Generation"
_NeurIPS.cc/2025/Conference — NeurIPS 2025 poster_

### Official Review · Reviewer_xn72 · 2025-06-30

**Clarity:** 2
**Significance:** 2
**Originality:** 2
**Rating:** 4
**Confidence:** 4

**Summary:**

The authors propose OmniGen-AR, a unified autoregressive framework capable of generating images from various input modalities, including text prompts, spatial conditions (e.g., segmentation masks, depth maps), and visual contexts (e.g., image editing, frame prediction, video generation). OmniGen-AR utilizes a shared visual tokenizer for discretizing various visual inputs and incorporates a separate text tokenizer for text prompts. A key challenge addressed is the tendency of autoregressive models to overfit to input images in editing tasks, leading to poor instruction adherence. To address this issue, the authors propose Disentangled Causal Attention (DCA), an attention mechanism that probabilistically masks the input image during the training phase. This prevents the model from excessively relying on the input image, thereby enhancing its ability to follow instructions accurately.

**Questions:**

Please refer to W1, W2, and W3.

**Ethical Concerns:**

["NO or VERY MINOR ethics concerns only"]

**Final Justification:**

I raised my rating from 3 to 4, as the rebuttal successfully addressed key concerns.

**Resolved Issues:**
- Provided comparisons with existing models
- Additional results on text-to-image, frame prediction, and depth-to-image
- Empirical comparison between DCA and classifier-free guidance
- Analysis of the discrepancy with ControlAR
- Analysis of DCA probability effects.

**Points that would further strengthen the paper:**
- Add direct experiments with EditAR beyond seg-to-img and depth-to-img, such as image editing and edge-to-image.
- Provide a qualitative comparison among OmniGen-AR, EditAR, and ControlAR.

**Limitations:**

yes

**Quality:**

2

**Strengths And Weaknesses:**

Strengths

S1. The authors tackle an important problem in image generation by proposing a unified framework that flexibly accommodates diverse input modalities.

S2. The authors provide comprehensive experiments revealing that autoregressive image generation models tend to overfit to input images, leading to poor instruction adherence.

Weakness

W1. The paper lacks comparison with existing models.
The paper lacks a thorough review and comparison with existing autoregressive image generation models. In particular, EditAR [1] also encodes multiple visual modalities using a unified visual tokenizer and employs an autoregressive transformer for conditional generation tasks, including diverse image-editing operations. The authors should cite this work and clarify the differences between OmniGen-AR and EditAR.


[1] Mu, Jiteng, Nuno Vasconcelos, and Xiaolong Wang. "EditAR: Unified Conditional Generation with Autoregressive Models." arXiv preprint arXiv:2501.04699 (2025).

W2. Experiments need to be improved.
W2.1. The authors should include EditAR[1] as a direct competitor, as mentioned in W1, to support their claims on performance and generality.
W2.2. Table 5 reports results only on text-to-video, image-editing, and mask-to-image tasks, omitting key tasks such as text-to-image, frame prediction, and depth-to-image. These omissions weaken the claim that DCA improves robustness across diverse generation tasks by reducing over-reliance on segmentation inputs.
W2.3. Although the authors describe conceptual differences between DCA and classifier-free guidance [2], they do not provide any empirical comparison, making it difficult to assess the effectiveness of the proposed technique.


[2] Ho, Jonathan, and Tim Salimans. "Classifier-Free Diffusion Guidance." NeurIPS 2021 Workshop on Deep Generative Models and Downstream Applications.

W3. The experimental analysis seems incomprehensive.
W3.1. In Table 4 (right), ControlAR [3], which is cited as the most relevant baseline, outperforms the proposed OmniGen-AR on both mask-to-image and depth-to-image tasks. However, the authors do not address or explain this discrepancy, making it unclear what advantages OmniGen-AR offers.
W3.2. In Table 6, the authors attribute performance degradation in T2I and T2V tasks under joint training to the “lower visual quality of editing and spatial-conditioned datasets,” but providing actual sample outputs would help validate this claim.
W3.3. In Table 5, the effect of varying DCA probabilities is reported, but the underlying trade-offs (e.g., why performance drops after 10%) are not analyzed.

[3] Li, Zongming, et al. "Controlar: Controllable image generation with autoregressive models." arXiv preprint arXiv:2410.02705 (2024).

---

> ### Author Rebuttal · Authors · 2025-07-30
>
> **Q1: Comparison with existing models**
> A1: Thank you for pointing this out. We appreciate the reviewer bringing EditAR to our attention and agree that it is relevant work that should be cited and discussed. We will include a citation and clarify the distinctions in the revised version of the paper:
>
> While EditAR and our method both employ autoregressive transformers and support multiple conditional image generation tasks, they are fundamentally different in two aspects: 1) EditAR is specifically designed for image editing and low-level control tasks (e.g., depth-to-image, edge-to-image, segmentation-to-image), while OmniGen-AR is a unified Any-to-Image framework that handles a broader range of input modalities. 2) EditAR aims to improve text-image alignment by introducing distillation loss, while our model hopes to prevent information leakage through DCA, a novel training-time attention mechanism that disentangles condition and content attention paths.
>
> **Q2: More results on text-to-image, frame prediction, and depth-to-image**
> A2: Thanks! We will add more quantitative results in the revised paper:
>
> | Method | T2I | VBench | FP  | Emu-CT | Mask | Depth |
> |--------|-----|--------|-----|--------|------|--------|
> | 0%     | 0.53 | 70.33 | 779 | 0.15   | 24.76 | 30.35 |
> | 5%     | 0.54 | 74.55 | 682 | 0.17   | 25.78 | 31.57 |
> | 10%    | 0.55 | 74.72 | 613 | 0.20   | 25.33 | 32.94 |
> | 20%    | 0.52 | 73.28 | 610 | 0.21   | 25.16 | 31.64 |
> | 30%    | 0.54 | 71.69 | 628 | 0.19   | 21.49 | 29.31 |
>
> **Q3: Empirical comparison between  DCA and classifier-free guidance**
> A3: Thanks for your great suggestion! The proposed DCA is fundamentally different from CFG, as it does not directly drop the conditions. Instead, it calibrates the attention between condition and content tokens to allow the model better disentangle and leverage conditional information. Below we show empirical comparison between them. As can be seen, compared to CFG, DCA leads to significantly better results on tasks like frame prediction, editing, and controllable generation.
>
> | Method | T2I  | VBench | FP  | Emu-CT | Mask | Depth |
> |--------|------|--------|-----|--------|------|--------|
> | DCA    | 0.55 | 74.72  | 613 | 0.20   | 25.33| 32.94  |
> | CFG    | 0.56 | 72.63  | 854 | 0.17   | 23.43| 27.98  |
>
> **Q4: Analysis of the discrepancy with ControlAR**
> A4: We acknowledge that ControlAR achieves higher mIoU on both mask-to-image and depth-to-image tasks. However, it’s important to highlight that ControlAR is specifically designed for controllable generation and relies on an additional control encoder (initialized from DINOv2-S), which is pretrained for dense visual understanding. This encoder introduces strong spatial priors that benefit structured control tasks but are task-specific and less generalizable to other modalities.
>
> In contrast, OmniGen-AR maintains a unified architecture without any task-specific heads or encoders, supporting a wider variety of inputs, including text, segmentation, depth, canny edges, image editing, and video frame prediction, all using a single causal transformer. While we may not outperform ControlAR on a few specific spatial tasks, we offer greater flexibility, simplicity, and generality, which are essential for Any-to-Image generation.
>
> **Q5: Visualization of editing and spatial-conditioned datasets**
> A5: Thanks! We will add this to our revised paper.
>
> **Q6: Analysis of the effects of DCA probabilities**
> A6: We thank the reviewer for raising this point. While DCA helps prevent condition-content leakage during training, applying it too frequently disrupts the model’s ability to fully leverage causal dependencies, especially when condition tokens carry rich information (e.g., dense segmentation or image context). In our experiments, 10% DCA strikes a balance: it encourages robust conditioning without significantly limiting the model’s access to informative context. We will update the paper to include more detailed analysis.

---

> > ### Comment · Reviewer_xn72 · 2025-08-04
> >
> > Thank you for your detailed responses addressing some of the key points raised in my initial review. While several concerns have been resolved, a few points remain unclear:
> >
> > Q1-1. Q4. While the rebuttal appears reasonable at first glance, several key claims remain insufficiently supported by evidence. For example, please list input modalities supported by OmniGen-AR but not by EditAR. The following claim is not fully substantiated: “offer greater flexibility, simplicity, and generality, which are essential for Any-to-Image generation.”
> >
> > Q1-2. As requested in W2.1, have you conducted direct performance comparisons with EditAR on overlapping tasks (e.g., image editing, depth-to-image, edge-to-image, segmentation-to-image)? Section 1 states that DCA not only prevents information leakage but also promotes more instruction-compliant predictions. Since both DCA in OmniGen-AR and the distillation loss in EditAR aim to enhance instruction compliance, such comparisons would clarify which model achieves stronger compliance.
> >
> > Q2. The additional results you reported seem inconsistent with those in the paper. Specifically, the DCA 10% results for FP, Emu-CT, and Depth are 613, 0.20, and 32.94, whereas Table 4 reports 429, 0.23, and 37.42. Could you clarify the cause of this discrepancy?

---

> > > ### Author Response · Authors · 2025-08-04
> > > **Response to Reviewer xn72**
> > >
> > > Thanks a lot for your feedback! We feel really glad that our rebuttal "addressed some of the key points raised in your initial review". Below we respond to your additional questions.
> > >
> > > **Q1-1. Q4. While the rebuttal appears reasonable at first glance, several key claims remain insufficiently supported by evidence. For example, please list input modalities supported by OmniGen-AR but not by EditAR. The following claim is not fully substantiated: “offer greater flexibility, simplicity, and generality, which are essential for Any-to-Image generation.”**
> > >
> > > A1-1: Our original saying is “While we may not outperform ControlAR on a few specific spatial tasks, we offer greater flexibility, simplicity, and generality, which are essential for Any-to-Image generation.” The statement was made in the context of comparing OmniGen-AR to ControlAR, instead of EditAR.
> > >
> > > In the rebuttal, we have discussed the discrepancy with ControlAR (Q4) and EditAR (Q1). Compared to ControlAR, we additionally support text modality (for text-to-image generation) and visual context (for image editing and video generation).
> > >
> > > **Q1-2. As requested in W2.1, have you conducted direct performance comparisons with EditAR on overlapping tasks (e.g., image editing, depth-to-image, edge-to-image, segmentation-to-image)? Section 1 states that DCA not only prevents information leakage but also promotes more instruction-compliant predictions. Since both DCA in OmniGen-AR and the distillation loss in EditAR aim to enhance instruction compliance, such comparisons would clarify which model achieves stronger compliance.**
> > >
> > > A1-2: Thanks for raising this. We completely agree that both DCA in OmniGen-AR and the distillation loss in EditAR are designed to enhance instruction compliance, and that a direct comparison helps clarify the effectiveness of these approaches.
> > >
> > > Below we compare the depth-to-image (on MultiGen) and segmentation-to-image (on COCOStuff) performance of both models. Since EditAR only reported the segmentation-to-image result on COCOStuff, we evaluate our model on the same benchmark to ensure fairness. The results demonstrate that even using fewer parameters, our model outperforms EditAR on both tasks, indicating that DCA is an effective and lightweight mechanism for improving conditional alignment in spatially grounded tasks, without requiring additional pretrained distillation targets
> > >
> > > | Method                                  | Mask (mIoU↑) | Depth (RMSE↓) |
> > > |-----------------------------------------|----------------|-----------------|
> > > | EditAR (initialized from LlamaGen-0.8B) | 22.62          | 34.93           |
> > > | Ours-0.5B                               | 23.73      | 32.94       |
> > >
> > > We will include this comparison in the revised version of the paper.
> > >
> > > **Q2. The additional results you reported seem inconsistent with those in the paper. Specifically, the DCA 10% results for FP, Emu-CT, and Depth are 613, 0.20, and 32.94, whereas Table 4 reports 429, 0.23, and 37.42. Could you clarify the cause of this discrepancy?**
> > >
> > > A2: As we mentioned in the paper, the reported results, 429, 0.23, and 37.42, are from 1.5B models, while 613, 0.20, and 32.94 are from 0.5B models.
> > >
> > > Here we would also like to discuss the reason why our 1.5B model performs worse than the 0.5B model on depth-to-image generation task. We hypothesize this may be due to the use of a large amount of depth-to-image training data, combined with the nature of depth signals, which provide dense low-level geometric constraints. Such signals can lead larger models to overfit or learn spurious patterns. The smaller 0.5B model with limited capacity, may capture the direct mapping from depth to image more stably under the current setup.

---

> > > > ### Comment · Reviewer_xn72 · 2025-08-05
> > > >
> > > > Thank you for your response. My concerns have been addressed.

---

> > > > > ### Author Response · Authors · 2025-08-05
> > > > > **Response to Reviewer xn72**
> > > > >
> > > > > Thank you very much for your response. We’re pleased to hear that your concerns have been addressed. We appreciate your time and thoughtful review, and hope our clarifications could contribute positively to your evaluation.

---

### Official Review · Reviewer_eCVT · 2025-07-01

**Clarity:** 2
**Significance:** 2
**Originality:** 2
**Rating:** 4
**Confidence:** 4

**Summary:**

This paper presents OmniGen-AR, a unified autoregressive framework for any-to-image generation that can handle diverse conditional inputs including text prompts, segmentation masks, depth maps, and reference images within a single model. The key technical innovation is the use of a shared visual tokenizer for different visual conditions, and a Disentangled Causal Attention (DCA) mechanism that aims to prevent information leakage between condition tokens and content tokens during training, improving instruction-following behavior. OmniGen-AR is evaluated on a wide variety of tasks (text-to-image, text-to-video, image editing, frame prediction, depth-to-image, segmentation-to-image) and demonstrates strong performance across established benchmarks, even surpassing diffusion-based competitors with comparable or fewer parameters.

**Questions:**

1. Can you provide a more formal or theoretical justification for DCA beyond the intuition of “blocking leakage”? For example, how does it relate to causal language modeling guarantees or to other attention regularizers?
2. In Section 4.4, you mention failures in removing objects or correctly following edits. Could you analyze why these errors happen — is it data sparsity, model capacity, or the autoregressive formulation? This would help readers understand the model’s true limitations.
3. AR models are generally slower than diffusion models for image synthesis due to token-by-token decoding. Could you quantify the latency or token generation speed in practice, compared to diffusion models? This would be important for practitioners considering adoption.
4. The results on segmentation-conditioned tasks are slightly worse than ControlAR or OmniGen. Do you have insights on how your approach might be adapted to improve geometric consistency in such spatially precise tasks?
5. The paper mentions failure cases, but are there potential misuse or bias issues with a general-purpose any-to-image generator? Please consider adding more discussion on this front.

**Ethical Concerns:**

["NO or VERY MINOR ethics concerns only"]

**Final Justification:**

I thank the authors for the reply, and part of my concerns are resolved, hence I raise my rating accordingly. However, I agree with other reviewers' comments that the quality and novelty of the paper is yet to be improved.

**Limitations:**

Yes, it's discussed.

**Quality:**

2

**Strengths And Weaknesses:**

Strengths:
1. The paper shows a solid empirical evaluation over many tasks, with quantitative comparisons to both AR and diffusion baselines, as well as reasonable ablation studies.
2. The writing is clear, with well-motivated design choices (e.g., why DCA is needed), and reasonable diagrams.
3. The work demonstrates that autoregressive models can be scaled to unified, multi-conditional image and video generation, which is significant given most recent focus on diffusion models.
4. While unifying multiple conditions is not a new direction, applying a shared visual tokenizer with disentangled causal attention within an AR framework is novel and technically interesting.

Weaknesses:
1. The empirical evaluation, although extensive, still shows weaker results on some metrics (e.g., segmentation-to-image mIoU is worse than ControlAR/OmniGen) and does not fully analyze why.
2. The treatment of failure cases is rather brief; the analysis in Section 4.4 is minimal, leaving unclear how to address the observed editing failures.
3. The DCA mechanism is an elegant solution, but the rationale and its theoretical justification are a bit ad hoc, relying on intuition rather than rigorous justification.
4. The paper lacks deeper discussion on training cost, sampling latency, or practical deployment of a large AR model for real-time tasks, which would help judge its feasibility in practice.

---

> ### Author Rebuttal · Authors · 2025-07-30
>
> **Q1: Weaker results on some metrics and further analysis**
> A1: Thanks for pointing this out! We acknowledge that our model shows weaker performance on segmentation-to-image in terms of mIoU compared to ControlAR and OmniGen, and will add more analysis in the revised paper.
> - For ControlAR, it leverages a dedicated control encoder initialized from DINOv2-S, which provides strong representations tailored for visual understanding. In contrast, our model uses a unified causal transformer without condition-specific encoders, aiming for architectural simplicity and generality across a wide range of tasks—not only spatial control but also text, video, and editing-based generation. Moreover, ControlAR is limited to controllable generation tasks, while our framework supports a broader set of inputs, including text-to-image, image editing, depth, segmentation, and even text-to-video.
> - For OmniGen, the performance advantage stems in large part from the scale and diversity of its training data. OmniGen is trained on a massive self-constructed dataset, covering a wide spectrum of tasks including text-to-image (~80M), image editing, human motion, virtual try-on, style transfer, subject-driven generation (e.g., GRIT-Entity, Web Images), and structured control generation (e.g., LAION with depth, segmentation, pose, and canny annotations, RefCOCO, ADE20k, and ReasonSeg). In comparison, our model is trained on a more modest dataset: ~56M T2I samples and ~13M for editing and control generation, using only publicly available sources.
> Despite these differences, our model still achieves strong and competitive results across most benchmarks, while maintaining a clean, unified autoregressive architecture that handles diverse input modalities without task-specific heads or encoders. We believe this highlights the scalability and generality of our approach.
>
> **Q2: The analysis in Section 4.4 is minimal, leaving unclear how to address observed editing failures**
> A2: Thank you for pointing this out. We will expand Section 4.4 in the revised version to include a more detailed analysis of failure cases: As shown in Figure 8, failure cases can be broadly categorized into two types: 1) Instruction-following capability. For instance, in the first row of Figure 8, the instruction is “Remove the bag on the bench next to the person sitting at the bus stop”, but the model removes the person instead. This indicates a failure in grounding fine-grained spatial and referential cues from language into visual modifications. 2) Low-quality generations under sparse control signals. Examples in the second row (depth-to-image and segmentation-to-image) show blurry or structurally inconsistent results, which likely stem from noisy supervision and sparse training coverage for these conditions.
> These failure modes suggest two potential directions for future work: 1) Scaling up the model and training data to build a stronger base model with improved generalization and instruction-following ability across diverse visual tasks. 2) Leveraging chain-of-thought (CoT) to improve the model’s reasoning ability on complex prompts.
>
> **Q3: The rationale of DCA mechanism**
> A3: Thank you for your comment. Disentangled Causal Attention (DCA) is designed to address a concrete and critical issue in conditional autoregressive generation: the risk of information leakage from condition tokens to content tokens during training. In standard causal attention, the model may inadvertently learn to exploit trivial correlations or positional cues between condition and target tokens, rather than learning meaningful conditional generation.
>
> DCA introduces a principled modification to the attention mask, splitting it into condition and content causal regions. This separation enforces a clearer dependency structure, encouraging the model to learn robust alignment between inputs and outputs without relying on token overlap or order-based shortcuts. While DCA is simple in form, it is grounded in a clear rationale: to preserve the autoregressive training signal while regularizing the attention pathway in a way that reflects the asymmetric role of condition versus content tokens. Its effectiveness is consistently validated across multiple conditional tasks, as shown in our ablation results, and it plays a key role in enabling our model to generalize across diverse inputs.
>
> **Q4: Discussion on training cost, sampling latency, or practical deployment of a large AR model**
> A4: Thanks for your valuable suggestions! We will include the discussion on training cost, sampling latency, and practical deployment in our revised paper:
> - Training cost: We train our model on 64 A100 GPUs. The 512 pretraining of 0.5B/1.5B models took 6/10 days separately, while the 1024 sft stage took 4/6 days.
> - Sampling latency and practical deployment: below we show that the AR visual generative model is compatible with the vLLM framework and deploying with it could significantly improve the inference speed.
>
> | Method           | # of Tokens | Speed (Sec/Image) |
> |------------------|-------------|-------------------|
> | SANA (1.6B)      | 32×32       | 1.77              |
> | Ours (1.5B)      | 32×32       | 43.55             |
> | Ours+vLLM (1.5B) | 32×32       | 5.40              |

---

> ### Author Response · Authors · 2025-08-05
> **Looking forward to your discussion**
>
> Dear reviewer, thank you again for your insightful comments on our paper, and we genuinely hope that our response could address your concerns. As the discussion is about to end, we are sincerely looking forward to your feedback. Please feel free to contact us if you have any further inquiries.

---

> > ### Comment · Reviewer_eCVT · 2025-08-05
> >
> > I thank the authors for the reply, and part of my concerns are resolved, hence I raise my rating accordingly.

---

> > > ### Author Response · Authors · 2025-08-06
> > > **Response to Reviewer eCVT**
> > >
> > > Thanks a lot for your positive feedback and raising your rating accordingly!

---

### Official Review · Reviewer_SePx · 2025-07-02

**Clarity:** 4
**Significance:** 4
**Originality:** 4
**Rating:** 5
**Confidence:** 5

**Summary:**

This paper introduces OmniGen-AR, the first unified autoregressive (AR) model that can generate images from a wide variety of inputs like text, segmentation maps, or video frames, all within a single framework. Its success comes from two key ideas: a shared tokenizer that turns all visual inputs into a common language, and a special training technique called Disentangled Causal Attention (DCA) that stops the model from simply copying the input. The model achieves state-of-the-art results on numerous tasks, proving that AR models are a powerful alternative to diffusion models for universal image generation.

**Questions:**

1. What is the total amount of training data used by the model? A brief comparison with other methods is needed.
2. Is it too straightforward to mask out the condition? Is it possible to use loss constraints in a way that prevents information leakage?
3. Are there more quantitative experimental comparisons between the 1.5B model and the 0.5B model?

**Ethical Concerns:**

["NO or VERY MINOR ethics concerns only"]

**Limitations:**

See Questions

**Quality:**

4

**Strengths And Weaknesses:**

Streangths:
1. It is the first work to systematically develop and validate a unified autoregressive framework for such a broad range of conditional image generation and understanding tasks.
2. The authors astutely identify "information leakage" as a novel and crucial challenge in a unified AR setting. The proposed Disentangled Causal Attention (DCA) is an elegant and effective solution.
Weaknesses:
1. The ablation study (Table 6) commendably reports that multi-task training, while beneficial for editing and spatial control, slightly degrades performance on the core text-to-image task. This indicates the presence of negative transfer or task interference, an important issue that is not fully resolved. It suggests that simply mixing all tasks together may not be the optimal strategy and that more advanced multi-task learning techniques might be necessary.
2. There are still some meaningful ablation experiments that can be conducted.

---

> ### Author Rebuttal · Authors · 2025-07-30
>
> **Q1: Total amount of training data and a brief comparison with other methods**
> A1: Thanks for your suggestion, below we compare the training data with several baselines:
>
> | Method         | Unified | T2I  | T2V  | Editing                                                                                     | ControlGen                                     |
> |----------------|---------|------|------|---------------------------------------------------------------------------------------------|------------------------------------------------|
> | **LLamaGen** | ❎      | Laion (50M) + Internal (10M) | -    | -                                                                                           | -                                              |
> | **OmniGen**        | ✅      | DataComp(56M) + SA(11M) + Laion(4M) + ShareGPT4v(1.26M) + ALLaVA-4V(1M) + DOCCI(15) + DenseFusion(1M) + JourneyDB(4M) + Internal(16M) | - | MagicBrush + Instruct-Pix2Pix + SEED-Edit + SomethingSomething + HR-VITON + FashionTryon + StyleBooth | MultiGen |
> | **Unified-IO2**       | ✅      |  LAION-400M + CC3M + CC12M + RedCaps + OBELICS | YT-Temporal-1B + ACAV100M + AudioSet + WebVid-10M + HDVILA-10M + Ego4D | -                                              |
> | **EditAR**         | ✅      | initialized from LLamaGen  | -    | SEEDData-Edit-Unsplash(1.5M) + PIPE (1.8M), COCOStuff + MultiGen                         | -                                              |
> | **ControlAR**      | ✅      | initialized from LLamaGen | -    | -                                                                                           | LAION-Aesthetics, ImageNet + ADE20K + COCOStuff + MultiGen |
> | **Janus-Pro (1B)** | ❎      | Laion(12M) + ImageNet(1M) + SA(11M) + OpenImages(8M) + Megalith(8M) + YFCC(15M) + JourneyDB(4M) +  Dalle3-high-quality-captions(1M) + PixelProse(16M) + Internal(72M) | -    | -                                                                                           | -                                              |
> | **Show-O (1.3B)**  | ❎      | ImageNet(1.28M)+ CC12M + SA(11M) + Laion(35M) + DataComp + COYO(2B) + JourneyDB(4M) | -    | -                                                                                           | -                                              |
> | **Ours**           | ✅      | CC3M + CC12M + OpenImages(8M) + SA(11M) + Megalith(8M) + JourneyDB(4M) +  Dalle3-high-quality-captions(1M) + Internal(10M) | Panda + HD-VILA(9M) + OpenSora pexels(45k) + OpenVid-1M + Internal(0.5M)                      | MagicBrush + Instruct-Pix2Pix + SEED-Edit  | MultiGen |
>
> **Q2: Ablation studies on using loss constraints to prevent information leakage**
> A2: Great suggestion! We conducted experiments to apply loss reweighting between condition and content tokens—specifically, assigning a lower weight (0.1) to condition tokens and a standard weight (1.0) to content tokens during training. The results below show that the proposed DCA outperforms loss reweighting on all benchmarks by clear margins.
>
> | Method   | VBench | Emu-CT | Mask  |
> |----------|--------|--------|-------|
> | baseline | 70.33  | 0.15   | 24.76 |
> | DCA      | 74.72  | 0.20   | 25.33 |
> | Loss     | 72.54  | 0.14   | 24.94 |
>
> **Q3: Quantitative experimental comparisons between the 1.5B model and the 0.5B model**
> A3: Below we report more quantitative experimental comparisons between 0.5B and 1.5B models, 1.5B model consistently achieve better results than 0.5B model.
>
> | Model Size | FramePred | Emu-CT | Mask |
> |------------|--------------------|------------|----------|
> | 0.5B       | 613                | 0.20       | 25.33    |
> | 1.5B       | 429                | 0.23       | 35.28    |

---

### Official Review · Reviewer_EJMS · 2025-07-02

**Clarity:** 2
**Significance:** 2
**Originality:** 2
**Rating:** 4
**Confidence:** 4

**Summary:**

The paper introduce a unified autoregressive model for any-to-image generation. The input for the model is a discrete sequence of text and condition image (depth/semantic map), then, generate a sequence of discrete visual tokens. In addition, the model is capable of generating videos from text. The author also propose disentangle causal attention to mitigate information leakage.

**Questions:**

- The paper mentioned "adopt Qwen2.5 as the text tokenier and transformer mode", does that mean the author finetune a pretrained Qwen2.5 for image generation or initialized a model with similar architecture as Qwen2.5 and trained it from scratch? If the author finetuned  a pretrained model, how does it perform on natural language understanding tasks after "fine-tuning"? If the model is trained from scratch, what is the reasoning of not using pre-trained weights?
- What is the inference speed of the introduced model? how is it compared to AR and Diffusion/Flow matching model of same scale?

**Ethical Concerns:**

["NO or VERY MINOR ethics concerns only"]

**Final Justification:**

The author addressed most of my concerns in the rebuttal so I slightly increase my score

**Limitations:**

Yes

**Quality:**

2

**Strengths And Weaknesses:**

## Strengths
- The paper is well written and easy to follow.
- The experiment and evaluation is fairly extensive.

## Weaknesses:
- The author claim the introduced model is the first AR for Any-to-Image generation, which is incorrect. There are several attempts that train an autoregressive image generative model for both text and images as input [1,2,3].
- The author claim to achieve state-of-the-art on benchmarks, which doesn't seem to be true. For instance, their model gets (substantial) lower performance compared to several models of the same size both AR [4], Diffusion [5], and hybrid model [6].
- The novelty of the paper is limited. The author simply extend the conventional next-token prediction on latent from VQ-VAE. Thus, I don't see how the model can tackle the inherited problem of previous AR/next-token image generation models e.g., slow inference speed and unidirectional generation.  The novelty is from the DCA, which is borderline in my point of view.
- For me, the main purpose of using AR image generation over diffusion or VAR is the simplicity when implementing join training with language modeling task. This may enable several useful properties such as in-context learning and complex language instruction. The paper doesn't demonstrate any convincing reasons for using AR. In fact, the performance and speed seems to be worse than diffusion/flow matching.


[1] Unified-IO: A Unified Model for Vision, Language, and Multi-Modal Tasks

[2] Unified-IO 2: Scaling Autoregressive Multimodal Models with Vision, Language, Audio, and Action

[3] Lumina-mGPT: Illuminate Flexible Photorealistic Text-to-Image Generation with Multimodal Generative Pretraining

[4] Janus-Pro: Unified Multimodal Understanding and Generation with Data and Model Scaling

[5] SANA: Efficient High-Resolution Image Synthesis with Linear Diffusion Transformers

[6] Show-o: One Single Transformer to Unify Multimodal Understanding and Generation

---

> ### Author Rebuttal · Authors · 2025-07-30
>
> **Q1: The claim of "the first AR for Any-to-Image generation"**
> A1: Thank you for the valuable comment. We agree that there have been prior autoregressive models that support multiple input modalities, including text and image. However, our goal is to develop a unified autoregressive visual generation model that explicitly accommodates three types of conditions, text, spatial, and visual contex, in a single framework. While models like Unified-IO and Unified-IO 2 indeed support text and spatial inputs, they do not explicitly unify visual context as a generative condition. Lumina-mGPT is trained with diverse inputs, but to the best of our knowledge, it only reports performance on text-to-image generation, without demonstrating generalization to the broader range of input conditions explored in our work. We will revise our claim in the paper to better reflect this nuance.
>
> **Q2: The claim of "achieving state-of-the-art on benchmarks"**
> A2: Thank you for your comment. We will revise the paper to more accurately characterize our results in comparison to prior work, below we would like to provide some important context regarding the comparisons mentioned: 1) diffusion models like [5] typically rely on additional text encoders that are typically not counted in the model size. For example, [5] uses a strong 2B-parameter text encoder (Gemma-2B), which is non-negligible compared to the diffusion models themselves. In contrast, our model uses a single causal transformer without external text encoders, keeping the architecture clean and parameter-efficient. 2) AR model [4] and hybrid model [6] are trained on much more data than our model. Specifically, our model is trained on approximately 56M text-to-image pairs and 13M examples for image editing and control generation. In contrast, [4] reports using around 140M samples, while [6] uses roughly 2B examples. Despite this significant parameter or data gap, our model achieves competitive performance across a range of benchmarks, especially given our unified support for diverse conditional generation tasks.
>
> **Q3: Novelty of this work**
> Thank you for the thoughtful comments. We would like to clarify the novelty and motivation of our work: we do not seek to alter the autoregressive modeling manner, but to build a unified AR framework that accommodates diverse conditional inputs. To alleviate the information leakage issue from condition tokens to content tokens in causal modeling, we introduce Disentangled Causal Attention (DCA), a training-time regularization scheme that carefully preserves the causal nature of AR generation while enabling the model to learn condition-aware generation without overfitting or shortcutting. Though lightweight by design, DCA is proved to be effective in improving the instruction following capability of our model, as evidenced by our ablation studies.
>
> **Q4: The advantage of using AR for image generation**
> Thanks for pointing this out. Our choice of an autoregressive (AR) model is motivated by its ability to flexibly accommodate diverse input conditions within a single causal transformer, in addition to the advantage you noted. There is currently no clear consensus in the community regarding which family of generative models, AR, diffusion, or VAR, is fundamentally superior to visual generation. Each has its own trade-offs in terms of quality, extensibility, and inference speed. We believe AR models remain an important and under-explored direction, especially in the context of multi-conditional visual generation, and our work aims to push this frontier forward.
>
> **Q5: Qwen initialization**
> Thank you for the question. We fine-tune a pretrained Qwen2.5 model for image generation tasks. Specifically, we adopt Qwen2.5 both as the text tokenizer and as the initialization for the transformer decoder. During training, we compute the autoregressive loss on the entire sequence, including both the text tokens (input prompts) and the visual tokens (generated image latents).
>
> |               | GenEval | MMMU |
> |---------------|---------|------|
> | Qwen2.5-0.5B  | - | 47.5 |
> | Pretrain w/o init | 0.29    | -    |
> | Pretrain w/ init  | 0.31    | 12.7 |
>
> - Performance with and without Qwen initialization: we conducted experiments using both Qwen-initialized and randomly initialized models and found that both setups can achieve similar performance on text-to-image (T2I) benchmarks after convergence (0.29 vs 0.31 on GenEval after 512 resolution pretraining). However, Qwen initialization leads to more stable training in the early stages, with lower loss.
> - Natural language understanding performance after T2I fine-tuning: after fine-tuning Qwen2.5 on T2I data, we observe a significant drop in language understanding performance, with MMLU accuracy decreasing from 47.5 to 12.7. This degradation is expected as we do not include any text-only data during fine-tuning. Although we compute loss on text tokens, the learning signal is relatively simple and repetitive, and lacks the diversity and complexity of language modeling tasks. We believe that the text understanding capability can be better preserved by mixing in text-only data for joint training, and we leave a systematic study of its impact on both language and T2I performance for future work.
>
> **Q6: Inference speed**
> A6: Thank you for the question. We evaluate the inference speed of our model alongside other models of comparable scale. Since Janus only supports 384×384 resolution, we only report its speed based on 24×24 tokens. As shown below, benefiting from the usage of KV cache, the inference speed of our model clearly outperforms Show-o. While our generation speed is still slower than SANA, deploying with vLLM could significantly narrow the gap.
>
> | Method              | # of Tokens | Speed (Sec/Image) |
> |---------------------|-------------|--------------------|
> | Janus (1.5B)        | 24×24       | 11.92              |
> | Show-o (1.3B)       | 32×32       | 269.10             |
> | SANA (1.6B)         | 32×32       | 1.77               |
> | Ours (1.5B)         | 32×32       | 43.55              |
> | Ours + vLLM (1.5B)  | 32×32       | 5.40               |

---

> ### Author Response · Authors · 2025-08-05
> **Looking forward to your discussion**
>
> Dear reviewer, thank you again for your insightful comments on our paper, and we genuinely hope that our response could address your concerns. As the discussion is about to end, we are sincerely looking forward to your feedback. Please feel free to contact us if you have any further inquiries.

---

> ### Author Response · Authors · 2025-08-07
> **Looking forward to your discussion**
>
> Hi Reviewer EJMS, since the reviewer-authors discussion time window will be closed soon, can you help read the rebuttal as soon as possible so that we can address any remaining concerns you may have. Grateful for your effort!

---

> > ### Comment · Reviewer_EJMS · 2025-08-07
> > **Reviewer's Response**
> >
> > Thanks for the detailed rebuttal, most of my concerns are addressed. I'll adjust my score accordingly. Minor detail but it will be clearer if the author can also report GPU model and whether or not tensor parallel is used for the inference speed table.

---

> > > ### Author Response · Authors · 2025-08-07
> > > **Response to Reviewer EJMS**
> > >
> > > Thanks for your reply! We evaluated all the models on Nvidia A100 GPU and tensor parallel was not used.

---

### Note · Authors · 2025-08-13

We sincerely appreciate the assistance from the area chair for managing the review process, and thank all the reviewers for their detailed and constructive comments. We are encouraged that the reviewers find the paper is **well-written** (Reviewer EJMS, eCVT), the proposed method is **elegant and effective** (Reviewer SePx), **novel and interesting** (Reviewer eCVT), **tackle an important challenge** (Reviewer SePx and xn72), and the experiments are **extensive** (Reviewer EJMS and xn72).

Based on reviewers' valuable comments, we conducted additional experiments and clarifications to address their concerns, including:

- [Reviewer EJMS]: we clarified several statements, discussed the novelty of this paper, the advantages of AR models for visual generation, and analyzed the effects of model initialization and inference speed using experimental results.
- [Reviewer SePx]: we listed the training data used and provided additional ablation study results.
- [Reviewer eCVT]: we analyzed the differences from related works, including ControlAR and OmniGen, discussed the rationale behind the DCA design, and the training and inference costs of our method.
- [Reviewer xn72]: we provided additional ablation study results (DCA v.s. CFG), discussed the differences from existing methods, and committed to presenting more visualization results in the revised paper.

We feel glad that according to the reviewers‘ responses, we have already addressed most of their concerns. We will ensure that all suggested revisions and clarifications are incorporated into the final version.

---

### Decision · Program_Chairs · 2025-09-17

**Decision:**

Accept (poster)

**Comment:**

This paper proposes OmniGen-AR, a unified AR framework for any-to-image generation that can take as input diverse conditions including text prompts, segmentation masks, depth maps, and reference images within a single model. The key technical innovation is the use of a shared visual tokenizer for different visual conditions, and a Disentangled Causal Attention (DCA) mechanism to prevent information leakage between condition tokens and content tokens during training, improving instruction-following behavior.

The strengths of the work include clear motivation (unified AR model with any input), moderate novelty (shared visual tokenizer with DCA), elegant solution (DCA), solid empirical evaluations, and clear writing.
Some negative aspects include some weaker results on some metrics, some unaddressed failure cases, lack of theoretical justification, lack of computational cost analysis, etc., most of which are solved by the author rebuttal. Therefore, all the reviewers are finally satisfied with the work and give Accept (x1) and Borderline Accept (x3). Based on the above analysis, the AC think it deserves an Accept.